# Motion-aware Latent Diffusion Models for Video Frame Interpolation

### Zhilin Huang
Shenzhen International Graduate
School, Tsinghua University
Pengcheng Laboratory
Shenzhen, China
zerinhwang03@pku.edu.cn

### Yijie Yu
Shenzhen International Graduate
School, Tsinghua University
Pengcheng Laboratory
Shenzhen, China
yyj23@mails.tsinghua.edu.cn

### Ling Yang
Peking University
Beijing, China
yangling0818@163.com

### Chujun Qin
China Southern Power Grid
Guangzhou, China
chujun.qin@pku.edu.cn

### Bing Zheng
Shenzhen International Graduate
School, Tsinghua University
Pengcheng Laboratory
Shenzhen, China
zhengb21@mails.tsinghua.edu.cn

### Xiawu Zheng
Xiamen University
Pengcheng Laboratory
Xiamen, China
zhengxiawu@xmu.edu.cn

### Zikun Zhou*
Pengcheng Laboratory
Shenzhen, China
zhouzikunhit@gmail.com

### Yaowei Wang
Pengcheng Laboratory
Harbin Institute of Technology
Shenzhen, China
wangyw@pcl.ac.cn

### Wenming Yang*
Shenzhen International Graduate
School, Tsinghua University
Pengcheng Laboratory
Shenzhen, China
yangelwm@163.com

## ABSTRACT

With the advancement of AIGC, video frame interpolation (VFI) has become a crucial component in existing video generation frameworks, attracting widespread research interest. For the VFI task, the motion estimation between neighboring frames plays a crucial role in avoiding motion ambiguity. However, existing VFI methods always struggle to accurately predict the motion information between consecutive frames, and this imprecise estimation leads to blurred and visually incoherent interpolated frames. In this paper, we propose a novel diffusion framework, **M**otion-**A**ware latent **Diff**usion models (**MADiff**), which is specifically designed for the VFI task. By incorporating motion priors between the conditional neighboring frames with the target interpolated frame predicted throughout the diffusion sampling procedure, MADiff progressively refines the intermediate outcomes, culminating in generating both visually smooth and realistic results. Extensive experiments conducted on benchmark datasets demonstrate that our method achieves state-of-the-art performance significantly outperforming existing approaches, especially under challenging scenarios involving dynamic textures with complex motion.

*Corresponding author.

## CCS CONCEPTS

• **Computing methodologies** → **Reconstruction**.

## KEYWORDS

Generative Models, Diffusion Models, Video Frame Interpolation

**ACM Reference Format:**
Zhilin Huang, Yijie Yu, Ling Yang, Chujun Qin, Bing Zheng, Xiawu Zheng, Zikun Zhou, Yaowei Wang, and Wenming Yang. 2024. Motion-aware Latent Diffusion Models for Video Frame Interpolation. In *Proceedings of the 32nd ACM International Conference on Multimedia (MM '24), October 28-November 1, 2024, Melbourne, VIC, AustraliaProceedings of the 32nd ACM International Conference on Multimedia (MM'24), October 28-November 1, 2024, Melbourne, Australia.* ACM, New York, NY, USA, 10 pages. https://doi.org/10.1145/3664647.3680846

## 1 INTRODUCTION

Video frame interpolation (VFI) aims to generate intermediate frames between two consecutive video frames. It is commonly used to increase the frame resolution, e.g., to produce slow-motion content [29]. VFI has also been applied to video compression [69], video generation [21] and animation production [57].

Existing VFI methods [29, 31, 37, 48, 71] are mostly based on deep neural networks. The early deep learning methods always rely on 3D convolution [28] or RNNs [9] to model the contextual correlations of conditional neighboring frames. Inspired by the success of generative adversarial networks (GANs) in image synthesis, more recent attempts have been made to develop video frame interpolation methods by incorporating the adversarial loss. Owing to the remarkable generative capabilities of GANs, these methods exhibit significant efficacy in predicting interpolated video frames that

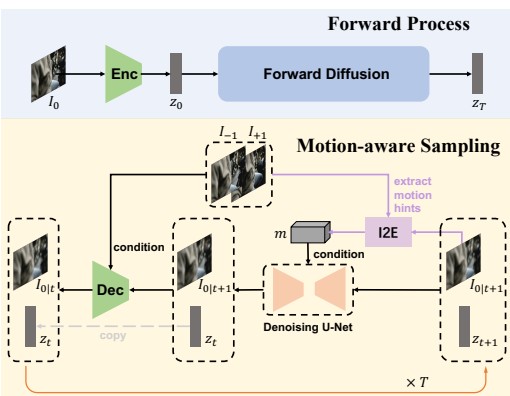

**Figure 1: Overview of the diffusion processes in MADIFF. The encoder and decoder enable projection between image and latent spaces, and the diffusion processes take place in the latent space.** *I2E denotes image-to-event generator* [78] **which have capability of generating event volume by taking two continuous frames as input. And** $m$ **denotes motion hints extracted from** *I2E.*

possess natural and consistent content, holding the state-of-the-art in the literature. However, these deep learning-based methods for VFI always tend to generate unrealistic texture, artifacts and low-perceptual results. The reason is that the primary contributor to the optimization objective — and consequently the final performance of the model — remains the L1/L2-based distortion loss between their outputs and the ground-truth interpolated frames [15]. This type of loss may not accurately reflect the perceptual quality of the interpolated videos. Consequently, [14] indicates that current techniques, despite attaining elevated PSNR scores, often fall short in terms of perceptual quality, particularly in demanding situations characterized by dynamic textures and complex motion patterns.

Recently, de-noising diffusion probabilistic models (DDPMs) have been gaining widespread interest and have risen to prominence as state-of-the-art solutions across various areas [20, 21, 23, 24, 61, 72]. These diffusion models have demonstrated exceptional capabilities in creating realistic and perceptually-enhanced images and videos, reportedly outperforming other generative models. [15, 67] are among the early methods that have explored the application of diffusion models for VFI tasks. Specifically, they address the VFI tasks as a form of conditional image generation by taking neighboring frames into the de-noising network for the target interpolated frame generation.

However, these methods fail to explicitly model the inter-frame motions between the interpolated frames and the given neighboring conditional frames, a crucial factor in preventing the generation of blurred interpolated frames due to motion ambiguity. This is particularly important in complex dynamic scenes that involve intricate motions, occlusions, or abrupt changes in brightness.

To tackle these challenges, in this paper, we propose a novel latent diffusion framework, **M**otion-**A**ware latent **Diff**usion model (MADIFF) for the video frame interpolation task. Specifically, we follow [15] to build upon MADIFF by adopting recently proposed latent diffusion models (LDMs) [55]. LDMs consist of an autoencoder that maps images into a latent space and a de-noising U-net, which

carries out the reverse diffusion process within that latent space, forming the foundation of our framework. To incorporate the inter-frame motion priors between given conditional neighboring frames with the interpolated frames into MADIFF, we propose a novel vector quantized motion-aware generative adversarial network, named VQ-MAGAN. In particular, we initially utilize a pre-trained Event-GAN [78] to predict the event volume which reflects the pixel-level intensity changes between two consecutive frames. Subsequently, the event volumes between the interpolated frame and the two neighboring conditional frames are employed as motion hints to enhance image reconstructions within the decoder of VQ-MAGAN. In this design, VQ-MAGAN possesses the capability to predict the target interpolated frame by aggregating contextual details from the given neighboring frames under the guidance of inter-frame motion hints. Furthermore, for the de-noising process in LDMs, we also incorporate motion hints between the interpolated frame and the two neighboring frames as additional conditions.

During the training process of both VQ-MAGAN and the de-noising U-net, we directly utilize the ground-truth interpolated frame for extracting inter-frame motion hints between interpolated frame and neighboring conditional frames. Since the ground-truth interpolated frame is unknown during the sampling process of LDM, extracting motion hints between the interpolated frame and the conditional neighboring frames is not feasible. To eliminate the discrepancy of motion hints extraction between the sampling phase and the training phase, making the motion hints in the sampling process available, we propose a novel motion-aware sampling procedure (MA-SAMPLING). Specifically, during the sampling process, we use the coarse interpolated frame predicted in the previous time step to extract inter-frame motion hints in conjunction with the conditional neighboring frames. The extracted motion hints are then fed into both VQ-MAGAN and the de-noising U-net for the prediction of the interpolated frame in the current time step. By refining the interpolated frames progressively, our MADIFF can effectively integrate the inter-frame motion hints into the sampling process, resulting in visually smooth and realistic frames.

Extensive experiments on various VFI benchmark datasets, encompassing both low and high resolution content (up to 4K), demonstrate that our MADIFF achieves the state-of-the-art performance significantly outperforming existing approaches, especially under challenging scenarios involving dynamic textures with complex motion. Our contributions are summarized as follows:

- We propose a novel vector quantized motion-aware generative adversarial network, named **VQ-MAGAN**, which fully incorporates the inter-frame motion hints between the target interpolated frame and the given neighboring conditional frames into the prediction of the interpolated frame.
- We propose a novel motion-aware sampling procedure, named **MA-SAMPLING**, to eliminate the discrepancy of motion hints extraction between the sampling phase and the training phase, making the extraction of motion hints in the sampling process feasible and refine the predicted interpolated frames progressively.
- We demonstrate, through quantitative and qualitative experiments, that the proposed method achieves the state-of-the-art performance outperforming existing approaches.

## 2 RELATED WORK

### 2.1 Image-to-Event Generation

Event cameras are a novel bio-inspired asynchronous sensor [18, 41] with advantages such as high dynamic range, high temporal resolution, and low power consumption [4, 18, 41]. Event cameras have the potential to provide solutions for a wide range of visually challenging scenarios and has already found extensive applications [2, 5, 22, 33, 38–40, 46, 54, 63, 64, 70, 77]. However, acquiring event stream data is expensive and requires specific devices. Recently, several studies have exploited the event simulation from the standard camera, i.e. simulating the event stream from continuous images or video sequences. Kaiser et al. [30] generate events by simply applying a threshold on the image difference. A positive or negative event is generated depending on the pixel's intensity difference. Pix2NVS [3] computes per-pixel luminance from conventional video frames. The technique generates synthetic events with inaccurate timestamps clustered to frame timestamps. [76, 78] propose a GAN-based method to generate the event volume by simply taking two continuous image frames as input. By modeling the correlation between continuous frames and event volumes, image-to-event generation models can capture inter-frame motion hints, which can serve as auxiliary guidance for the VFI task. In our MADIFF, we directly utilized the pre-trained EventGAN [78] to extract motion hints for guiding the interpolation process.

### 2.2 Video Frame Interpolation

Current approaches to video frame interpolation (VFI) that leverage deep learning can typically be divided into two main categories: those that are based on flow estimation and those that rely on kernel methods. Existing flow-based methods [29, 35, 42, 44, 47, 48, 50, 51, 56, 71] primarily generate intermediate frames by analyzing the pixel motion between consecutive frames. Specifically, these methods use optical flow estimation to capture the movement of objects within the scene, and then synthesize intermediate frames based on these estimations. These methods can be founded on various assumptions and algorithms, such as assuming uniform motion within the scene or employing machine learning models to predict the optical flow field, thereby enhancing the accuracy and visual quality of the interpolated frames. On the other hand, Existing kernel-based video frame interpolation methods [7, 8, 11, 16, 37, 49] typically utilize kernel functions to estimate the relationships between pixels across different frames, generating intermediate frames by performing a weighted average of pixel values based on known frames. These methods can flexibly adapt to various motion patterns in the scene by selecting appropriate kernel sizes and shapes, thus effectively smoothing the visual transitions between frames while preserving details. In addition to these two main categories, there are also some methods integrate flow-based and kernel-based approaches to better synthesis performance [10, 31].

Meanwhile, several works attempt to introduce the event stream as a supplemental conditions for guiding accurate models between the neighboring frames and interpolated frames. However, almost event-based methods require taking the accurate ground-truth event streams as inputs which is inconvenient for practical application. In contrast, our MADIFF directly utilizes off-the-shelf pre-trained image-to-event models for providing the motion hints

in the subsequent interpolated frame generation process. Besides, our MADIFF refines the target interpolated frame in a progressive manner, which is quite different from previous VFI methods that predict interpolated frame in a one-shot manner.

Recently, denoising diffusion probabilistic models (DDPMs) have gained increasing attention and emerged as a new state-of-the-art in several areas of computer vision. [15, 67] are the most early methods exploiting the application of DDPMs for VFI tasks. Compared to previous VFI methods, diffusion-based methods show satisfactory perceptual performance, especially on dynamic textures with complex motions. However, existing diffusion-based methods simply define the VFI task as a conditional image generation task by considering the neighboring frames as conditions, neglecting to explicitly model inter-frame motions which are a crucial factor for generating realistic and visually smooth interpolated frames. Compared with these methods, we propose a novel diffusion framework, MADIFF, to effectively incorporate motion hints from existing motion-related models into diffusion models in a progressive manner.

## 3 PRELIMINARY

### 3.1 Representation of Event Volume

Each event $\mathbf{e}$ can be represented by a tuple $(x, y, t, p)$, where $x$ and $y$ represent the spatial position of the event, $t$ represents the timestamp, and $p = \pm 1$ indicates its polarity.

As described in [78], the event are presented for easily processing: events are scattered into a fixed size 3D spatio-temporal volume, where each event, $(x, y, t, p)$ is inserted into the volume, which has $B = 9$ temporal channels, with a linear kernel:

$$t_i^* = (B - 1) \cdot \frac{t_i - t_1}{t_N - t_1} \tag{1}$$

$$V(x, y, t) = \sum_i max(0, 1 - |t - t_i^*|) \tag{2}$$

This retains the distribution of the events in $x$-$y$-$t$ space, and has shown success in a number of tasks [6, 53, 79].

Since the event volume generated by EventGAN [78] are the concatenation of event volumes of different polarity along the time dimension, the final event volume which is strictly non-negative. In our MADIFF, we directly utilize the event volume generated by EventGAN as the inter-frame motion hints.

### 3.2 Latent Diffusion Models

Latent diffusion models (LDMs) [55] is variant of de-noising diffusion probabilistic models which executes the de-noising process in the latent space of an autoencoder, namely $\mathcal{E}(\cdot)$ and $\mathcal{D}(\cdot)$, implemented as pre-trained VQ-GAN [17] or VQ-VAE [65]. Compared with executing de-noising process in the pixel-level data, LDM can reduce computational costs while preserving high visual quality.

For the training of LDM, the given latent code $z$ for a randomly sampled training image $x$ is converted to noise with a Markov process defined by the transition kernel:

$$q(z_t \mid z_{t-1}) = \mathcal{N}(z_t; \sqrt{\alpha_t} z_{t-1}, (1 - \alpha_t)\mathbf{I}) \tag{3}$$

where $t = 1, 2, \cdots, T$, $z_0 = z$, and $\alpha_t$ is a hyper-parameter that controls the rate of noise injection. When the amount of noise is sufficiently large, $z_T$ becomes approximately distributed according

to $\mathcal{N}(0, \mathbf{I})$. In order to convert noise back to data for sample generation, the reverse diffusion process is estimated by learning the reverse transition kernel:

$$p_\theta(z_{t-1} \mid z_t) = \mathcal{N}(\mu_\theta(z_t), \Sigma_\theta(z_t)) \tag{4}$$

and then take it as an approximation to $q(z_{t-1} \mid z_t)$. Following Ho et al. [20], $\mu_\theta(z_t)$ is parameterized with a neural network $\epsilon_\theta(z_t, t)$ (called the score model [59, 60]) and $\Sigma_\theta$ is fixed to be a constant. The score model can be optimized with de-noising score matching [26, 66]. During sample generation, within each time step, the de-noising U-net initially predicts the $\hat{z}_0$. Finally, the decoder of VQ-GAN or VQ-VAE generates the image $\hat{I}_0$ from the de-noised latent representation $\hat{z}_0$, disregarding any contextual information.

## 4 METHODS

### 4.1 Motion Hints Extraction

In MADiff, we utilize the pre-trained EventGAN [78] to capture inter-frame motion hints between the interpolated frame with the conditional neighboring frames. Specifically, given the interpolated frame $I_0 \in \mathbb{R}^{H \times W \times 3}$ and two conditional neighboring frames $I_{-1}, I_{+1} \in \mathbb{R}^{H \times W \times 3}$, where $I_{-1}$ denotes the previous frame and $I_{+1}$ denotes the next frame. The motion hints extraction process are formulated as follows:

$$m_{-1 \to 0} = f_{I2E}(I_{-1}, I_0) \tag{5}$$

$$m_{0 \to +1} = f_{I2E}(I_0, I_{+1}) \tag{6}$$

where $f_{I2E}(\cdot)$ is the pre-trained EventGAN, $m_{i \to j}$ denotes the extracted motion hints from the frame $i$ to the frame $j$. In practice, we directly use the predicted event volume $EV_{i \to j} \in \mathbb{R}^{H \times W \times (2 \times B)}$ as $m_{i \to j}$. Besides, MADiff we proposed is a general framework which can incorporate different motion-related models easily. More details please refer to Section 5.5.2.

### 4.2 VQ-MAGAN

*Implementation Details.* For the consideration of motion information between the interpolated frame and neighboring frames is crucial for the VFI task, we propose a novel VQ-GAN, namely VQ-MAGAN as presented in Figure 2.

The encoder $\mathcal{E}$ produces the latent encoding $z_0 = \mathcal{E}(I_0)$ by taking the given ground-truth target frame $I_0 \in \mathbb{R}^{H \times W \times 3}$ as the input, where $z_0 \in \mathbb{R}^{\frac{H}{f} \times \frac{W}{f} \times 3}$, and $f$ is a hyper-parameter. In practice, we set $f = 32$ following [15].

Then the decoder $\mathcal{D}$ reconstruct target frame $\hat{I}_0$ by taking $z_0$ and the feature pyramids $\phi_{-1}, \phi_{+1}$ which are extracted by $\mathcal{E}$ from two neighboring frames $I_{-1}, I_{+1}$ following [15]. Moreover, we utilize the motion hints extractor to capture the inter-frame motion hints $m_{-1 \to 0}$ and $m_{0 \to +1}$ between the ground-truth target frame $I_0$ with the neighboring frames $I_{-1}$ and $I_{+1}$. And then we take the $m_{-1 \to 0}$ and $m_{0 \to +1}$ as an additional guidance for the layer-wisely contextual aggregation in the decoder $\mathcal{D}$ through a motion-aware warp (MA-Warp) module. Specifically, for the feature $h_0^l \in \mathbb{R}^{U \times V \times C}$ of interpolated frame and the motion hints $m_{-1 \to 0}, m_{0 \to +1} \in \mathbb{R}^{H \times W \times (2 \times B)}$, the MA-Warp in $l$-th decoder layer firstly reshape the motion hints to the resolution of $U \times V$, obtained $m_{-1 \to 0}^l, m_{0 \to +1}^l \in \mathbb{R}^{U \times V \times (2 \times B)}$. Then for each motion hints, a 2-channel offset map $\Omega_{-1 \to 0}^l$ and

$\Omega_{0 \to +1}^l$, respectively, which reflects the pixel-level feature correlations from the neighboring frame to the target interpolated frame is generated through a learnable neural networks: $f_\Omega(\cdot)$

$$\Omega_{0 \leftarrow -1}^l = f_\Omega(h_0^l, m_{-1 \to 0}^l, \phi_{-1}^l) \tag{7}$$

$$\Omega_{0 \leftarrow +1}^l = f_\Omega(h_0^l, m_{0 \to +1}^l, \phi_{+1}^l) \tag{8}$$

After that, a warp function $f_{warp}(\cdot)$ proposed in [25] is introduced and serves as a aggregation mechanism:

$$h_{0 \leftarrow -1}^l = f_{warp}(\Omega_{0 \leftarrow -1}^l, \phi_{-1}^l) \tag{9}$$

$$h_{0 \leftarrow +1}^l = f_{warp}(\Omega_{0 \leftarrow +1}^l, \phi_{+1}^l) \tag{10}$$

The MA-Warp also generates a gate map $g \in [0, 1]^{U \times V \times 1}$ to account for occlusion [29], and a residual map $\delta \in \mathbb{R}^{U \times V \times C}$ to further enhance the performance:

$$\tilde{h}_0^l = g \cdot h_{0 \leftarrow -1}^l + (1 - g) \cdot h_{0 \leftarrow +1}^l + \delta, \tag{11}$$

$$g = f_g(h_{0 \leftarrow -1}^l, h_{0 \leftarrow +1}^l), \tag{12}$$

$$\delta = f_\delta(h_0^l). \tag{13}$$

where both $f_g(\cdot)$ and $f_\delta(\cdot)$ are learnable neural networks, $\tilde{h}_0^l$ is the output of MA-Warp in $l$-th decoder layer. By hierarchically applying MA-Warp in the decoder layer, VQ-MAGAN is able to fully utilize the motion hints for accurately aggregating pyramid contexts from neighboring frames. Compared with VQ-FIGAN proposed in [15], our VQ-MAGAN has capability of incorporating the inter-frame motions between the target interpolated frames with the conditional neighboring frames.

*Training VQ-MAGAN.* For the training of VQ-MAGAN, we follow the original training settings of VQGAN in [17, 55], where the loss function consists of an LPIPS-based [75] perceptual loss, a patch-based adversarial loss [27] and a latent regularization term based on a vector quantization (VQ) layer [65]. Particularly, we use the ground-truth interpolated frame to extract motion hints with given conditional neighboring frames.

Since the ground-truth target frame are provided for extracting motion hints during the training process of VQ-MAGAN, the reconstruction task may becomes much easier for VQ-MAGAN, potentially degrading the performance of reconstruction in the inference stage. To address this issue, we only utilize the motion hints with a probability of 0.5 during the training stage to assist in the reconstruction process of VQ-MAGAN. The pseudo code is provided in the Appendix. While we replace the ground-truth interpolated frame with the predicted one from the previous time step during the sampling procedure as described in Section 4.4.

### 4.3 De-noising with Conditional Motion Hints

*Implementation Details.* The encoder of trained VQ-MAGAN allows us to access a compact latent space in which we perform forward diffusion by gradually adding Gaussian noise to the latent $z_0$ of the target frame $I_0$ according to a pre-defined noise schedule, and learn the reverse (de-noising) process to perform conditional generation. Moreover, compared with previous methods, we additionally incorporate the dynamic inter-frame motion hints into the de-noising process. To this end, we adopt the noise-prediction parameterization [20] of diffusion models and train a de-noising U-Net by

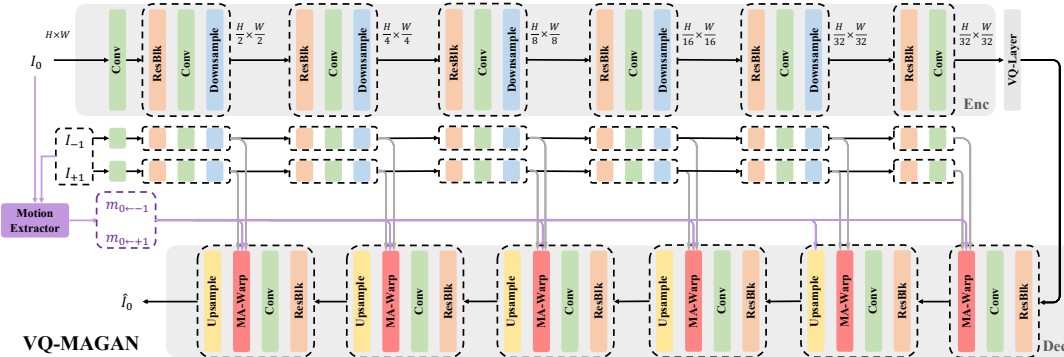

**Figure 2: The architecture of the vector quantized motion-aware generative adversarial network, VQ-MAGAN. In practice, the motion extractor is image-to-event generator [78], $m_{i \to j}$ denotes the inter-frame motion hints between frame $i$ and $j$.**

minimizing the re-weighted variational lower bound on the conditional log-likelihood $\log p_\theta(z_0 | z_{-1}, z_{+1}, m_{-1 \to 0}, m_{0 \to +1})$, where $z_{-1}$ and $z_{+1}$ are the latent representations of the two conditional neighboring frames, and $m_{i \to j}$ denotes the motion hints between frame $i$ and $j$ as described in Section 4.1.

Specifically, the de-noising U-Net $\epsilon_\theta$ takes as input the noisy latent representation $z_t$ for the target frame $I_0$ (sampled from the $t$-th step in the forward diffusion process of length $T$), the diffusion step $t$, as well as the conditioning latent representations $z_{-1}, z_{+1}$ for the neighboring frames $I_{-1}, I_{+1}$. It is trained to predict the noise added to $z_0$ at each time step $t$ by minimizing

$$\mathcal{L} = \mathbb{E}\big[\, \|\epsilon - \epsilon_\theta(z_t, t, z_{-1}, z_{+1}, m_{-1 \to 0}, m_{0 \to +1})\|^2 \,\big] \quad (14)$$

where $t \sim \mathcal{U}(1, T)$. The derivation and full details of the training procedure of $\epsilon_\theta$ are provide in Appendix. Intuitively, the training is performed by alternately adding a random Gaussian noise to $z_0$ according to a pre-defined noising schedule, and having the network $\epsilon_\theta$ predict the noise added given the step $t$, conditioning on $z_{-1}, z_{+1}$ and $m_{i \to j}$.

*Training of De-noising U-net.* It is worth noting that during the training of de-noising U-net, we directly utilize the ground-truth interpolated frame for extracting inter-frame motion hints which is the same as the extraction process in the training of VQ-MAGAN.

### 4.4 MA-Sampling of MADiff

As described above, both VQ-MAGAN and de-noising U-net are conditioned on the inter-frame motion hints extracted from the interpolated frame and the conditional neighboring frames. During the training stage of both VQ-MAGAN and de-noising U-net we directly using the ground-truth interpolated frame for the motion hints extraction in a teach-forcing manner. However, the interpolated frame is unknown in the sampling phase, rendering the extraction of inter-frame motion hints between the interpolated frame and neighboring frames infeasible. While the motion hints directly extracted from given neighboring frames are often inaccurate and cannot provide sufficient guidance, leading to sub-optimal performance as described in Table 5. For eliminating this discrepancy of motion hints extraction between the training and sampling

phase, making the motion hints in the sampling process available, we propose a novel MA-Sampling.

Before introducing MA-Sampling, we provide a review of the sampling process within traditional LDM for VFI tasks [15]: Within each time step, firstly the de-noising U-net $\epsilon_\theta$ predicts the noise $\hat{\epsilon}$ conditioned on the latent representations $z_{-1}, z_{+1}$ of neighboring frames $I_{-1}, I_{+1}$. Then $\hat{z}_{0|t}$ is obtained as follows:

$$\hat{\epsilon} = \epsilon_\theta(\hat{z}_t, t, z_{-1}, z_{+1}) \quad (15)$$

$$\hat{z}_{0|t} = \frac{1}{\sqrt{\alpha_t}}\big(\hat{z}_t - \frac{1 - \alpha_t}{\sqrt{1 - \bar{\alpha}_t}}\hat{\epsilon}\big) \quad (16)$$

where $\epsilon_\theta(\cdot)$ is the de-noising U-net, $\hat{z}_{0|t}$ denotes the predicted $z_0$ at time step $t$ (particularly, we denote $\hat{z}_{0|1}$ as $\hat{z}_0$ for simplification), $\hat{z}_t$ is the noisy latent representation of the predicted $\hat{z}_{0|t+1}$ obtained at previous time step $t + 1$ during the sampling process. And $\hat{z}_t$ can be calculated by using $\hat{\epsilon}$ and relevant parameters of the pre-defined forward process as eq (3). Finally, the decoder of VQ-GAN produces the image $\hat{I}_0$ from $\hat{z}_{0|1}$ with the help of feature pyramids $\phi_{-1}, \phi_{+1}$ extracted by the encoder $\mathcal{E}$ from $I_{-1}, I_{+1}$. Compared with the traditional sampling process mentioned above, our MA-Sampling has capability of incorporating accurate motion hints between the interpolated frame and the neighboring frames for progressively refining the predicted target frame. Specifically, at time step $t$, firstly the de-noising U-net $\epsilon_\theta$ predicts the noise $\hat{\epsilon}$ conditioned on the latent representations $z_{-1}, z_{+1}$ of neighboring frames and additional motion hints $\hat{m}_{0 \to +1|t+1}, \hat{m}_{-1 \to 0|t+1}$. Then $\hat{z}_{0|t}$ is obtained as follows:

$$\hat{\epsilon} = \epsilon_\theta(\hat{z}_t, t, z_{-1}, z_{+1}, \hat{m}_{-1 \to 0|t+1}, \hat{m}_{0 \to +1|t+1}) \quad (17)$$

$$\hat{z}_{0|t} = \frac{1}{\sqrt{\alpha_t}}\big(\hat{z}_t - \frac{1 - \alpha_t}{\sqrt{1 - \bar{\alpha}_t}}\hat{\epsilon}\big) \quad (18)$$

where $\hat{m}_{-1 \to 0|t+1}, \hat{m}_{0 \to +1|t+1}$ are extracted from the predicted interpolated frame $\hat{I}_{0|t+1}$ and neighboring frames $I_{-1}, I_{+1}$:

$$\hat{I}_{0|t+1} = \mathcal{D}(\hat{z}_{0|t+1}) \quad (19)$$

$$\hat{m}_{-1 \to 0|t+1} = f_{I2E}(I_{-1}, \hat{I}_{0|t+1}) \quad (20)$$

$$\hat{m}_{0 \to +1|t+1} = f_{I2E}(\hat{I}_{0|t+1}, I_{+1}) \quad (21)$$

And $\hat{z}_{t-1}$ can be calculated using $\hat{\epsilon}$ and relevant parameters of the pre-defined forward process as sampling process of previous methods eq (3). Particularly, at time step $T$, motion hints

$\hat{m}_{-1\to0|T+1}$ and $\hat{m}_{0\to+1|T+1}$ are both replaced with empty features $\mathbf{O} \in \mathbb{R}^{H\times W\times(2\times B)}$. Finally, the decoder $\mathcal{D}$ produces the interpolated frame $\hat{I}_{0|1}$ (for simplification we denote it as $\hat{I}_0$) from the de-noised latent representation $\hat{z}_{0|t+1}$ i.e. $\hat{z}_0$, by fully considering feature pyramids $\phi_{-1}, \phi_{+1}$ extracted by the encoder $\mathcal{E}$ from $I_{-1}, I_{+1}$ as contexts under the guidance of motion hints $\hat{m}_{-1\to0|1}, \hat{m}_{0\to+1|1}$. Full details and pseudo code are provided in the Appendix.

## 5 EXPERIMENTS

### 5.1 Implementation Details

Regarding the diffusion processes, following [15], we adopt a linear noise schedule and a codebook size of 8192 for vector quantization in VQ-MAGAN. We sample from all baseline diffusion models with the DDIM [58] sampler for 200 steps. We also follow [15] to train the VQ-MAGAN using the Adam optimizer [34] and the de-noising U-net using the Adam-W optimizer [43], with the initial learning rates set to $10^{-5}$ and $10^{-6}$ respectively. Single NVIDIA V100 GPU were used for all training and evaluation.

### 5.2 Experimental Setup

*5.2.1 Datasets.* In our experiments, we strictly follow the dataset configuration specified in [15] for both training and evaluation. For training MADIFF, we utilize the widely adopted Vimeo90k dataset [71]. And additional samples from the BVI-DVC dataset [45] are adopted to better evaluate our VFI methods across a broader range of scenarios. The overall training set comprises 64,612 frame triplets from Vimeo90k-septuplets and 17,600 frame triplets from BVI-DVC, utilizing only the central three frames. For data augmentation, we randomly crop $256 \times 256$ patches and perform random flipping and temporal order reversing following [15]. For evaluation, MADIFF is validated on four widely recognized VFI benchmarks, including Middlebury [1], UCF-101 [62], DAVIS [52] and SNU-FILM [10]. The resolutions of these test datasets vary from $225 \times 225$ up to 4K, covering various levels of VFI complexity.

*5.2.2 Evaluation.* Following [15], we adopt a perceptual image quality metric LPIPS [75], FloLPIPS [12] for performance evaluation. These metrics exhibit a stronger correlation with human assessments of VFI quality in contrast to the traditionally utilized quality metrics, such as PSNR and SSIM [68]. We also evaluate FID [19] which measures the similarity between the distributions of interpolated and ground-truth frames. FID was previously used as a perceptual metric for video compression [74], enhancement [73] and colorization [32]. We also provide benchmark results based on PSNR and SSIM in the Appendix, noting that these are limited in reflecting the perceptual quality of interpolated content [14] and are therefore not the focus of this paper.

*5.2.3 Baseline Methods.* MADIFF was compared against 11 recent SOTA VFI methods, including BMBC [50], AdaCoF [37], CDFI [16], XVFI [56], ABME [51], IFRNet [35], VFIformer [44], ST-MFNet [13], FLAVR [31], MCVD [67] and LDMVFI [15]. It it noted that MCVD and LDMVFI are diffusion-based VFI method.

### 5.3 Quantitative Comparison

Table 1 shows the performance of the evaluated methods on the Middlebury, UCF-101 and DAVIS test sets. It can be observed that MADIFF consistently outperforms all the other VFI methods including both non-diffusion or diffusion-based methods. Moreover, we evaluate the performance on the four splits of the SNU-FILM dataset (the average motion magnitude increases from Easy to Extreme), as summarized in Table 2, which further demonstrates the superior perceptual quality of MADIFF, especially in the scenes including complex motions (SNU-FILM-Hard and Extreme). Notably, the performance of other diffusion-based VFI methods, namely MCVD and LDMVFI, are generally unsatisfactory, which suggests that simply defining the VFI task as a conditional image generation task, without explicitly modeling inter-frame motions, may not be adequate for generating realistic and visually smooth results. As presented in Table 1, the number of parameters in MADIFF is large. This is because we adopted the existing de-noising U-net [55] designed for generic image generation following [15] and introduced adapters for fusing motion hints into the VQ-MAGAN and de-noising U-net.

### 5.4 Qualitative Comparison

Figure 3 shows the comparison between example frames interpolated by MADIFF with the competing methods. We can observe that, non-diffusion-based methods (BMBC [50], VFIformer [44], IFRNet [35] and ST-MFNet [13]) tend to predict blurry results due to the L1/L2-based distortion loss. On the contrary, diffusion-based VFI method (LDMVFI [15]) is able to generate more realistic results, while it also produces several artifacts leads by motion ambiguity. By incorporating inter-frame motion hints as the guidance, our MADIFF has capability of predicting realistic results.

### 5.5 Ablation Study

In this section we experimentally validate and study the main components including the VQ-MAGAN and MA-SAMPLING in MADIFF.

*5.5.1 Effectiveness of VQ-MAGAN and MA-SAMPLING.* To validate the effectiveness of the proposed VQ-MAGAN and MA-SAMPLING, we conduct 4 experiments: (1) **Exp0**: without introducing the extracted motion hints from the pre-trained motion-related model into the VQ-MAGAN, and replacing the MA-SAMPLING with the original sampling procedure from LDMs; (2) **Exp1**: without introducing the extracted motion hints from the pre-trained motion-related model into the VQ-MAGAN, while still using MA-SAMPLING for extracting inter-frame motion hints as additional conditions of the de-nosing U-net; (3) **Exp2**: introducing the extracted motion hints from the pre-trained motion-related model into the VQ-MAGAN, while replacing MA-SAMPLING with original sampling procedure in LDMs; (4) **Exp3**: MADIFF equipped with both VQ-MAGAN and MA-SAMPLING. The experimental results are presented in Table 3. By comparing **Exp0**, **Exp1** and **Exp2**, we can observe that simply introducing MA-SAMPLING can only brings slightly performance improvements, while simply introducing motion hints to guide the contexts warping process in VQ-MAGAN can significantly improve the performance of MADIFF. Moreover, the results

| | Middlebury | | | UCF-101 | | | DAVIS | | | #P (M) |
|---|---|---|---|---|---|---|---|---|---|---|
| | LPIPS↓ | FloLPIPS↓ | FID↓ | LPIPS↓ | FloLPIPS↓ | FID↓ | LPIPS↓ | FloLPIPS↓ | FID↓ | |
| BMBC | 0.023 | 0.037 | 12.974 | 0.034 | 0.045 | 33.171 | 0.125 | 0.185 | 15.354 | 11.0 |
| AdaCoF | 0.031 | 0.052 | 15.633 | 0.034 | 0.046 | 32.783 | 0.148 | 0.198 | 17.194 | 21.8 |
| CDFI | 0.022 | 0.043 | 12.224 | 0.036 | 0.049 | 33.742 | 0.157 | 0.211 | 18.098 | 5.0 |
| XVFI | 0.036 | 0.070 | 16.959 | 0.038 | 0.050 | 33.868 | 0.129 | 0.185 | 16.163 | 5.6 |
| ABME | 0.027 | 0.040 | 11.393 | 0.058 | 0.069 | 37.066 | 0.151 | 0.209 | 16.931 | 18.1 |
| IFRNet | 0.020 | 0.039 | 12.256 | 0.032 | 0.044 | 28.803 | 0.114 | 0.170 | 14.227 | 5.0 |
| VFIformer | 0.031 | 0.065 | 15.634 | 0.039 | 0.051 | 34.112 | 0.191 | 0.242 | 21.702 | 5.0 |
| ST-MFNet | N/A | N/A | N/A | 0.036 | 0.049 | 34.475 | 0.125 | 0.181 | 15.626 | 21.0 |
| FLAVR | N/A | N/A | N/A | 0.035 | 0.046 | 31.449 | 0.209 | 0.248 | 22.663 | 42.1 |
| MCVD | 0.123 | 0.138 | 41.053 | 0.155 | 0.169 | 102.054 | 0.247 | 0.293 | 28.002 | 27.3 |
| LDMVFI | 0.019 | 0.044 | 16.167 | 0.026 | 0.035 | 26.301 | 0.107 | 0.153 | 12.554 | 439.0 |
| MADiff w/o MS | 0.016 | 0.034 | 13.649 | 0.024 | 0.032 | 24.677 | 0.098 | 0.143 | 11.764 | 447.8 |
| MADiff | 0.016 | 0.034 | 11.678 | 0.024 | 0.033 | 24.289 | 0.096 | 0.142 | 11.089 | 448.8 |

Table 1: Quantitative comparison of MADiff ($f = 32$) and 11 tested methods on Middlebury, UCF-101 and DAVIS. Note ST-MFNet and FLAVR require four input frames so cannot be evaluated on Middlebury dataset which contains frame triplets. For each column, we highlight the best result in red and the second best in blue.

| | SNU-FILM-Easy | | | SNU-FILM-Medium | | | SNU-FILM-Hard | | | SNU-FILM-Extreme | | |
|---|---|---|---|---|---|---|---|---|---|---|---|---|
| | LPIPS↓ | FloLPIPS↓ | FID↓ | LPIPS↓ | FloLPIPS↓ | FID↓ | LPIPS↓ | FloLPIPS↓ | FID↓ | LPIPS↓ | FloLPIPS↓ | FID↓ |
| BMBC | 0.020 | 0.031 | 6.162 | 0.034 | 0.059 | 12.272 | 0.068 | 0.118 | 25.773 | 0.145 | 0.237 | 49.519 |
| AdaCoF | 0.021 | 0.033 | 6.587 | 0.039 | 0.066 | 14.173 | 0.080 | 0.131 | 27.982 | 0.152 | 0.234 | 52.848 |
| CDFI | 0.019 | 0.031 | 6.133 | 0.036 | 0.066 | 12.906 | 0.081 | 0.141 | 29.087 | 0.163 | 0.255 | 53.916 |
| XVFI | 0.022 | 0.037 | 7.401 | 0.039 | 0.072 | 16.000 | 0.075 | 0.138 | 29.483 | 0.142 | 0.233 | 54.449 |
| ABME | 0.022 | 0.034 | 6.363 | 0.042 | 0.076 | 15.159 | 0.092 | 0.168 | 34.236 | 0.182 | 0.300 | 63.561 |
| IFRNet | 0.019 | 0.030 | 5.939 | 0.033 | 0.058 | 12.084 | 0.065 | 0.122 | 25.436 | 0.136 | 0.229 | 50.047 |
| ST-MFNet | 0.019 | 0.031 | 5.973 | 0.036 | 0.061 | 11.716 | 0.073 | 0.123 | 25.512 | 0.148 | 0.238 | 53.563 |
| FLAVR | 0.022 | 0.034 | 6.320 | 0.049 | 0.077 | 15.006 | 0.112 | 0.169 | 34.746 | 0.217 | 0.303 | 72.673 |
| MCVD | 0.199 | 0.230 | 32.246 | 0.213 | 0.243 | 37.474 | 0.250 | 0.292 | 51.529 | 0.320 | 0.385 | 83.156 |
| LDMVFI | 0.014 | 0.024 | 5.752 | 0.028 | 0.053 | 12.485 | 0.060 | 0.114 | 26.520 | 0.123 | 0.204 | 47.042 |
| MADiff w/o MS | 0.013 | 0.021 | 5.157 | 0.025 | 0.048 | 10.919 | 0.058 | 0.110 | 23.143 | 0.125 | 0.210 | 49.435 |
| MADiff | 0.013 | 0.021 | 5.334 | 0.027 | 0.049 | 11.022 | 0.058 | 0.107 | 22.707 | 0.118 | 0.198 | 44.923 |

Table 2: Quantitative comparison results on SNU-FILM which contains 4 subsets with different motion complexities (the average motion magnitude increases from Easy to Extreme). And VFIformer is not included because the GPU goes out of memory. For each column, we highlight the best result in red and the second best in blue.

| Exps | VQ-MAGAN | MA-Sampling | Middlebury | | | UCF-101 | | | DAVIS | | |
|---|---|---|---|---|---|---|---|---|---|---|---|
| | | | LPIPS↓ | FloLPIPS↓ | FID↓ | LPIPS↓ | FloLPIPS↓ | FID↓ | LPIPS↓ | FloLPIPS↓ | FID↓ |
| Exp0 | ✗ | ✗ | 0.022 | 0.043 | 17.202 | 0.027 | 0.036 | 25.578 | 0.113 | 0.157 | 12.250 |
| Exp1 | ✗ | ✔ | 0.020 | 0.037 | 11.632 | 0.028 | 0.037 | 26.885 | 0.109 | 0.153 | 11.558 |
| Exp2 | ✔ | ✗ | 0.016 | 0.034 | 13.649 | 0.024 | 0.032 | 24.677 | 0.098 | 0.143 | 11.764 |
| Exp3 | ✔ | ✔ | 0.016 | 0.034 | 11.678 | 0.024 | 0.033 | 24.289 | 0.096 | 0.142 | 11.089 |

Table 3: Ablation study of main components of MADiff. For each column, we highlight the best result in red and the second best in blue.

of **Exp3** demonstrate that simultaneously applying VQ-MAGAN and MA-Sampling can help MADiff achieve optimal performance.

*5.5.2 Influence of Different Types of Motion Hints.* In MADiff, we propose a novel framework to bridge the motion-related prediction models and the diffusion-based VFI models. To demonstrate the generalizations of MADiff, we conduct ablation study on the effect of different motion hints. Specifically, we utilize two types of motion hints for guiding the interpolated generation process

in VQ-MAGAN and MA-Sampling of MADiff: (1) **Flow-based motion hints**: flow maps between the interpolated frame with the neighboring frames estimated by the pre-trained FastFlowNet [36]; (2) **Event-based motion hints**: event volumes between the interpolated frame with the neighboring frames predicted by the pre-trained EventGAN [78] as described in Section 4.1.

Experimental results are presented in Table 4, where the baseline refers to the MADiff without incorporating motion hints in

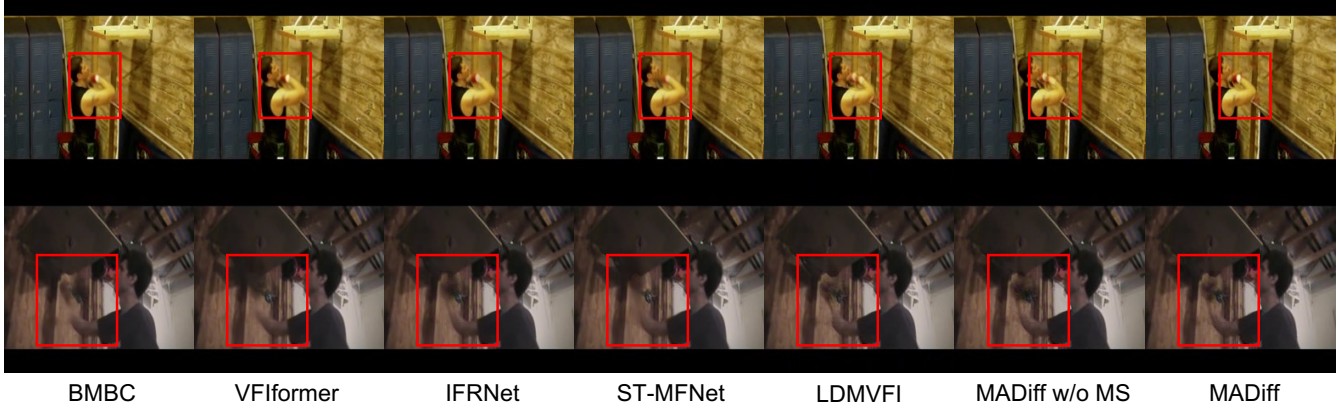

| | BMBC | VFIformer | IFRNet | ST-MFNet | LDMVFI | MADiff w/o MS | MADiff |

**Figure 3: Visual examples of frames interpolated by the state-of-the-art methods and the proposed MADIFF. Under large and complex motions, our method preserves the most high-frequency details, delivering superior perceptual quality.**

| Exps | Middlebury | | | UCF-101 | | | DAVIS | | |
|---|---|---|---|---|---|---|---|---|---|
| | LPIPS↓ | FloLPIPS↓ | FID↓ | LPIPS↓ | FloLPIPS↓ | FID↓ | LPIPS↓ | FloLPIPS↓ | FID↓ |
| baseline | 0.022 | 0.043 | 17.202 | 0.027 | 0.036 | 25.578 | 0.113 | 0.157 | 12.250 |
| Flow-based Motion Hints | 0.016 | 0.037 | 16.294 | 0.025 | 0.034 | 23.793 | 0.104 | 0.152 | 12.647 |
| Event-based Motion Hints | 0.016 | 0.034 | 11.678 | 0.024 | 0.033 | 24.289 | 0.096 | 0.142 | 11.089 |

**Table 4: Ablation study of different type of motion hints in MADIFF. For each column, we highlight the best result in red and the second best in blue.**

| Exps | Middlebury | | | UCF-101 | | | DAVIS | | |
|---|---|---|---|---|---|---|---|---|---|
| | LPIPS↓ | FloLPIPS↓ | FID↓ | LPIPS↓ | FloLPIPS↓ | FID↓ | LPIPS↓ | FloLPIPS↓ | FID↓ |
| baseline | 0.022 | 0.043 | 17.202 | 0.027 | 0.036 | 25.578 | 0.113 | 0.157 | 12.250 |
| Global Motion Hints | 0.018 | 0.034 | 11.990 | 0.025 | 0.034 | 23.757 | 0.104 | 0.150 | 11.865 |
| Dynamic Motion Hints | 0.016 | 0.034 | 11.678 | 0.024 | 0.033 | 24.289 | 0.096 | 0.142 | 11.089 |

**Table 5: Effectiveness of extracting motion hints between interpolated frames with neighboring frames instead of directly extracting motion hints from two continuous neighboring frames. For each column, we highlight the best result in red and the second best in blue.**

both VQ-MAGAN and MA-SAMPLING. The results indicate that our MADIFF can effectively integrate various types of motion hints, thereby enhancing the performance of video frame interpolation. This demonstrates that our MADIFF constitutes a flexible framework, allowing for the straightforward replacement of the motion hint extractor to incorporate diverse motion hints into diffusion models for the VFI task. Moreover, we conduct two additional experiments to ascertain the superiority of motion hints between the interpolated frame and the neighboring frames (referred to as **Dynamic Motion Hints** in Table 5) over motion hints between two neighboring frames (referred to as **Global Motion Hints** in Table 5). The results presented in Table 5 demonstrate that motion hints extracted between the interpolated frame and the neighboring frames offer more precise guidance and help to mitigate motion ambiguity in the VFI task, compared to motion hints directly extracted from two consecutive neighboring frames.

## 6 CONCLUSION

In this paper, for the VFI task, we propose a novel motion-aware latent diffusion model (MADIFF) which can fully leverage rich inter-frame motion priors from readily available and pre-trained motion-related models during the generation of interpolated frames. Specifically, our MADIFF consist of a vector quantized motion-aware generative adversarial network (VQ-MAGAN) and a de-noising U-net. And VQ-MAGAN is adept at aggregating contextual details under the guidance of inter-frame motion hints between the interpolated and given neighboring frames. Additionally, we propose a novel motion-aware sampling procedure (MA-SAMPLING) that progressively refines the predicted frame throughout the sampling process of the diffusion model. Comprehensive experiments conducted on benchmark datasets demonstrate that our MADIFF achieves the state-of-the-art performance, significantly surpassing existing approaches, particularly in scenarios characterized by dynamic textures and complex motions.

## ACKNOWLEDGMENTS

This work was partly supported by the National Natural Science Foundation of China (Nos.62171251&62311530100) and the Special Foundations for the Development of Strategic Emerging Industries of Shenzhen (No.KJZD20231023094700001) and the Major Key Research Project of PCL (No.PCL2023A08).

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
