# OpenReview forum: "Motion-aware Latent Diffusion Models for Video Frame Interpolation"
_acmmm.org/ACMMM/2024/Conference — MM2024 Poster_

### Official Review · Reviewer_r3yz · 2024-05-21

**Rating:** 5
**Confidence:** 3

**Summary:**

The paper introduces an LDM-based method for video frame interpolation, addressing the challenge of accurately predicting motion information between consecutive frames. The proposed method incorporates motion priors between neighboring frames and refines intermediate outcomes through a diffusion sampling procedure. Experiments show the effectiveness of the proposed method.

**Strengths:**

1.	The idea of introducing motion hints is interesting and will benefit the community.
2.	Extensive experiments demonstrate the effectiveness of proposed method.
3.	Ablation studies verify that each component is an optimal solution.

**Limitations:**

1.	Misleading and overstatement of contribution. The authors claim that the proposed method achieves SOTA performance in complex scenarios involving dynamic textures. However, this assertion is potentially misleading due to the omission of the VFITex dataset in their evaluation. VFITex is a widely used and challenging dataset specifically designed to test video frame interpolation methods on various dynamic textures. The absence of results on VFITex leads to an incomplete comparison, particularly with LDMVFI, which includes corresponding results on this dataset. Consequently, the claim of SOTA performance is difficult to verify and may mislead readers. It is recommended to include evaluations on the VFITex dataset to provide a more comprehensive and accurate comparison of performance.
2.	Difference with other motion-aware works. It is necessary to add a more thorough discussion for polishing clear motivation (i.e., comparison with existing motion-aware techniques in video generation). Incorporating motion information is a well-established approach in this field, as evidenced by previous works [1,2]. Additionally, there are several motion-aware diffusion models (DM) [3,4] that, while not specifically designed for video frame interpolation, are relevant and should be considered. The authors should discuss these methods in the introduction or related work sections to provide a clearer context for their contributions.
3.	Event across datasets. Leveraging event-based data is a promising approach due to its dense and efficient nature. However, there are concerns regarding the performance consistency of EventGAN across different datasets. EventGAN, trained on image pairs to simulate events for existing image datasets, may not perform uniformly well across all evaluated datasets. This inconsistency is evident in the results presented in Table 4, where the proposed components using motion hints do not consistently improve performance. For instance, the FID gain (with/without MA-sampling) is -0.046 for Middlebury and 0.469 for DAVIS, while for UCF101, the FID gain (with/without VQ-MAGAN) is 0.388. That would be far more interesting to analyze the inconsistency across datasets by some visualization and discussion. And this can motivate the following works.

[1] Decomposing Motion and Content for Video Generation, CVPR18

[2] MOSO: Decomposing Motion, Scene and Object for Video Prediction, CVPR23

[3] Distribution Extrapolation Diffusion Model for Video Prediction, CVPR24

[4] Conditional Image-to-Video Generation with Latent Flow Diffusion Models, CVPR23

**Suitability:**

3

---

### Official Review · Reviewer_orwT · 2024-05-24

**Rating:** 4
**Confidence:** 4

**Summary:**

This paper proposes MADIFF method to conduct video frame interpolation by incorporating motion priors among frames. The method establishes motion prior via a diffusion sampling procedure. However, additional clarification and comparisons may be needed for better understanding and evaluation.

**Strengths:**

1. This paper explores the denoising diffusion probabilistic model, a recent emerging technology, that is known for its ability to yield realistic and perceptually-optimized images.

**Limitations:**

1. In Figure 1, the mixed usage of subscripts with different meanings and no descriptions could introduce confusion. For example, the subscripts for I and z seem to have different meanings, i.e., some of them refer to frame index, while some of them refer to iterations of the diffusion process. Besides, this figure reminds me of Figure 1 of LDMVFI (AAAI 24), an earlier video frame interpolation method based on denoising diffusion probabilistic model. The proposed method is described as “significantly outperforming existing approaches”, but its qualitative results in Figure 6 seem to be very close to those of LDMVFI.
2. How to acquire z_t? Would the LDMs introduce randomness to the interpolation result? To clearly demonstrate the function of VQ-MAGAN in the diffusion process, is it possible to incorporate VQ-MAGAN in Figure 1?
3. How does the results compare to “Extracting motion and appearance via inter-frame attention for efficient video frame interpolation (CVPR 2023)”? The model size of the proposed model seems to be relatively large, e.g. 10 times larger compared with the existing methods.

**Suitability:**

2

---

### Official Review · Reviewer_rxpr · 2024-05-29

**Rating:** 4
**Confidence:** 2

**Summary:**

The paper presents a novel approach to video frame interpolation (VFI) using Motion-Aware Latent Diffusion models (MADiff). It aims to address issues of motion ambiguity and imprecise motion estimation in existing VFI methods, which often result in blurred and visually incoherent interpolated frames. The proposed MADiff framework incorporates motion priors between neighboring frames to refine intermediate outcomes progressively, achieving state-of-the-art performance on benchmark datasets, particularly in challenging scenarios involving dynamic textures with complex motion.

**Strengths:**

1) The introduction of motion-aware latent diffusion models is reasonable and somewhat innovative.
2) Extensive experiments conducted on various benchmark datasets, including high-resolution content up to 4K, demonstrate its performance improvements over the baselines.
3) The writing is good and the proposed method is detailed and logical.

**Limitations:**

1) There is a heavy reliance on quantitative metrics (LPIPS, FloLPIPS, FID) for evaluating performance. These metrics, while useful, do not fully capture perceptual quality, especially in complex motion scenarios. The paper could benefit from more diverse evaluation metrics, including user studies or perceptual quality assessments.
2) The proposed method appears to be computationally expensive due to the use of latent diffusion models and VQ-MAGAN.
3) The failure cases or limitations of the proposed method are suggested to be analyzed.

**Suitability:**

2

---

### Meta-Review · Area_Chair_SHeh · 2024-07-01

**Recommendation:** Accept (Poster)
**Confidence:** 4

**Metareview:**

This paper presented a method for video frame interpolation (VFI) that uses motion priors between conditional neighbouring frames in a diffusion framework. The proposed diffusion sampling process that establishes motion prior could be a contribution to the community. This paper received comments from three reviewers. The strengths of this work include: the proposed idea of the motion-aware diffusion models, extensive experiments (including ablation studies) to validate the effectiveness of the proposed method, and the writing. There are also some limitations: the evaluation of the quality of the results, unclear description of some claims and statements, computational complexity, and performance inconsistency across different data. All three reviewers recommended positive scores, with two borderline accept and one weak accept.

Considering the above points, the key idea of this work might be of interest to a group of audience in ACM MM and worth seeing and discussing. As a result, the AC recommends Accept.